# Evaluation of Complementary Feeding Indicators Among Children Aged 6–23 Months According to the Health Literacy Status of Their Mothers

**DOI:** 10.3390/nu16203537

**Published:** 2024-10-18

**Authors:** Sevim Gonca Kocagozoglu, Meltem Sengelen, Siddika Songul Yalcin

**Affiliations:** 1Department of Pediatrics, Faculty of Medicine, Kırıkkale University Hospital, 71450 Kırıkkale, Türkiye; 2Department of Social Pediatrics, Institute of Child Health, Hacettepe University, 06230 Ankara, Türkiye; 3Department of Public Health, Faculty of Medicine, Hacettepe University, 06230 Ankara, Türkiye; msengelen@yahoo.com; 4Division of Social Pediatrics, Department of Pediatrics, Hacettepe University İhsan Doğramacı Children’s Hospital, 06230 Ankara, Türkiye

**Keywords:** infant and young child feeding (IYCF), complementary feeding indicators, health literacy, TSOY-32

## Abstract

Background/Objectives: Infant and young child feeding (IYCF) practices directly affect child health, development, and survival, especially under 2 years of age and ultimately affect adult life well-being. As the primary caregivers of the children, mothers with higher health literacy may better perceive the benefits of optimal complementary feeding practices, leading to improved health outcomes for their children. In this study, we aimed to assess complementary feeding practices among children aged 6–23 months in Turkey according to 2021 World Health Organization IYCF indicators [minimum dietary diversity (MDD); minimum meal frequency (MMF); minimum acceptable diet (MAD); egg and/or flesh food consumption (EFF); sweet beverage consumption (SwB); unhealthy food consumption (UFC); zero vegetable or fruit consumption (ZVF); and bottle feeding (BoF)] and investigate their associations with sociodemographic characteristics and mothers’ health literacy. Methods: With a descriptive study design, we reached 572 mothers of children aged 6–23 months from five regions of Turkey. We used the Turkey Health Literacy Scale-32 (TSOY-32) to assess mothers’ health literacy. Results: While maternal and child age are significantly associated with more complementary feeding practices, specifically MDD, MAD, and EFF, having multiple children has negative impacts on several complementary feeding indicators, including MDD, MMF, MAD, UFC, and ZVF. The only indicator associated with mothers’ TSOY-32 scores was zero fruit and vegetable consumption. Conclusion: Raising awareness among mothers about the importance of complementary feeding practices and identification of vulnerable groups will guide practitioners and policymakers to improve child health and nutrition.

## 1. Introduction

Infants and young children (IYC) require adequate and balanced nutrition to ensure optimal growth and development. The complementary feeding period, which follows exclusive breastfeeding, is a critical phase in a child’s life. Nutritional inadequacies during this time can lead to growth failure, stunting, and increased risk of chronic non-communicable diseases such as obesity, hypertension, diabetes, and allergies, all of which can significantly impact adult health [1]. It has been shown that if populations can access evidence-based nutrition interventions, the total of deaths in children younger than 5 years can be reduced by 15% [2]. Therefore, the first 1000 days of life are the most critical period for a child to ensure optimal growth and development and to reach full capacity in adult life [3].

The World Health Organization (WHO) and the Turkish Ministry of Health advocate for exclusive breastfeeding for the first six months of life, followed by the introduction of solid, semi-solid, and soft foods after six months (180 days), with continued breastfeeding up to 24 months or more [4,5]. To support the growth, health, and behavioral development of IYC under two years, it is essential to assess and monitor their feeding practices both nationally and globally. In 2021, the WHO defined nine key complementary feeding indicators for breastfed and non-breastfed infants aged 6–23 months. These IYC feeding (IYCF) indicators for children aged 6–23 months (Figure 1) include the introduction of solid, semi-solid, or soft foods (ISSSF); minimum dietary diversity (MDD); minimum meal frequency (MMF); minimum acceptable diet (MAD); minimum milk feeding frequency for non-breastfed children (MMFF); egg and/or flesh food consumption (EFF); sweet beverage consumption (SwB); unhealthy food consumption (UFC); zero vegetable or fruit consumption (ZVF); and bottle feeding (BoF) [4].

Health literacy, defined as the capacity to obtain, process, and understand basic health information and services needed to make appropriate health decisions, plays a pivotal role in health outcomes [6]. Beyond basic reading ability, health literacy encompasses a holistic understanding of one’s health status, the ability to comprehend diagnoses, adherence to recommended health behaviors, and knowledge of how to effectively navigate the healthcare system [7]. As primary caregivers, mothers’ health literacy levels are closely linked to their children’s health outcomes. Research indicates that low health literacy in mothers correlates with more medication errors, increased emergency department visits, poor nutrition knowledge, poor parenting practices, higher obesity rates, and poorer asthma management in their children [8,9]. Notably, breastfeeding is the most extensively studied area within maternal health literacy and infant nutrition, with higher health literacy associated with improved rates of initiation and maintenance of breastfeeding, and also exclusive breastfeeding [10,11]. The literature reveals a lack of studies addressing health literacy in relation to complementary feeding. Only one study has examined maternal nutrition literacy and complementary feeding, demonstrating a correlation between a mother’s nutrition literacy level and both MMF and MAD [12]. This finding highlights a significant knowledge gap regarding the role of health literacy in shaping dietary behaviors, indicating that further research is required to explore this relationship more thoroughly and across diverse populations.

This study is grounded in the Health Belief Model (HBM), which posits that individual health behaviors are influenced by personal beliefs about health conditions, perceived benefits of action, and barriers to action [13]. Mothers with higher health literacy may better perceive the benefits of optimal complementary feeding practices, leading to improved health outcomes for their children. Additionally, the social cognitive theory (SCT) provides a framework for understanding how maternal education and socioeconomic status influence feeding behaviors through observational learning, social support, and self-efficacy [14]. These theoretical perspectives help explain how health literacy and sociodemographic factors interact to shape nutritional practices, ultimately impacting child health and development.

Therefore, we hypothesize that higher maternal health literacy will be positively associated with adherence to WHO-recommended complementary feeding practices, including greater dietary diversity and appropriate meal frequency. This study aims to evaluate complementary feeding practices among children aged 6–23 months in Turkey and explore their associations with maternal health literacy and sociodemographic characteristics. Understanding the association between maternal factors and IYCF practices could help policymakers design more effective public health strategies to support vulnerable populations.

## 2. Materials and Methods

### 2.1. Study Design

This is a descriptive study conducted between September and December 2023 via an online survey. We reached participants who were friends and followers of our social media accounts, and WhatsApp groups, using the formal snowball method [15].

### 2.2. Study Population

Mothers aged over 18 years who had children aged 6–23 months were included in the study. Mothers with psychiatric disorders were excluded. Infants whose care was not provided by the mother were excluded from the study. Mothers with at least primary school education were included in the study. Children born from multiple pregnancies were excluded*. Infants with neurodevelopmental disorders, chromosomal anomalies, and those requiring special diets due to phenylketonuria, celiac disease, and food allergies were excluded. If there were more than one baby aged 6–23 months, the survey was conducted for the older one.

Since there were no previous studies on health literacy and complementary feeding in the community, a preliminary study of 50 questionnaires was conducted to calculate the sample size. In this preliminary study, we found the rate of minimum dietary diversity to be 75%. Previously, women’s health literacy (sufficient and excellent scores) was reported as 27.5–47.5% (mean: 35.5) in Turkey [8,16]. According to the health literacy status of the mothers, allocation ratio (n2/n1) = 1.66 (64.5/35.5), p1 = 0.75, p2 = 0.60 (supposed 20% less in the group with inadequate health literacy), alpha error = 0.05 for two tail, power 95%, we calculated 551 mother–child pairs are necessary to predict differences in minimum dietary diversity (G*Power 3.1.9.4, Franz Faul, Universitata Kiel, Kiel, Germany). With a margin of error of 20%, the number of questionnaires to be reached was determined as 661.

### 2.3. Variables

The participants filled in a questionnaire with two parts. Part 1 included 22 questions about the sociodemographic characteristics of mothers and infants and the foods that the children ate the previous day (Appendix A). Part 2 included the Turkish Health Literacy Scale with 32 items (TSOY-32) (Appendix A).

The independent variables included the mother’s age (<30 years and ≥30 years), education level (≤12 years and >12 years), working status (house wife or working), birth place (according to 5 regions; north, west, east, central, south), TSOY-32 score (adequate–insufficient), family structure (nuclear–extended family), household income (income > outcome, income = outcome, income < outcome), single child status; children’s age (6–8 months, 9–11 months, 12–17 months, and 18–23 months), gender (male and female), and birth week (<38 weeks, ≥38 weeks).

We used the Turkey Health Literacy Scale-32 (TSOY-32) to assess mothers’ health literacy [16]. This scale is based on the conceptual framework developed by the European Health Literacy Research Consortium [17]. Each item was scored as 4 = very easy, 3 = easy, 2 = difficult, 1 = very difficult. A score of 0 was given for the expression “No opinion”. The calculation was made for those who answered at least 80% of the questions as follows: Formula = Index = (arithmetic mean − 1) × [50/3]. The level of health literacy was evaluated in four categories according to the score obtained: inadequate health literacy [0–25 points], problematic-limited health literacy [>25–33 points], adequate health literacy [>33–42 points], excellent health literacy [>42–50 points]. During the evaluation of complementary feeding indicators according to health literacy points, since sufficient numbers could not be reached with four categories we divided them into two groups: Inadequate and problematic health literacy groups were combined and named as “inadequate health literacy”; adequate and excellent health literacy groups were combined and named as “adequate health literacy”.

IYCF indicators was asked for dependent variables; CBF, MDD, MMF, MAD, EFF, SwB, UFC, ZVF, BoF (Figure 1) [4].

### 2.4. Data Analysis

The data were analyzed using IBM SPSS statistics software for Windows version 23.0 (Chicago, IL, USA). The distributions of categorical variables are given as numbers and percentages.

Chi-square test and binary logistic regression were used to determine the relationship between the independent and dependent variables. Binary logistic regression (Method: Enter) analyzed variables including maternal age (≥30 vs. <30 years), mother’s education year (>12 vs. ≤12 years), mother’s occupation (house wife vs. working), family structure (extended family vs. nuclear family), region (reference: West), number of children (>1 vs. 1), household income status (ref: income < outcome), maternal TSOY-32 score (adequate score vs. inadequate score), child’s age (reference: 6–8 months), baby birth week (≥38 vs. <38) for association with IYCF indicators. Adjusted odds ratio [AOR] with 95% confidence interval (CI) were calculated. In the regression analysis, multicollinearity among variables was not an issue, as all variance inflation factor (VIF) values were below 10 and tolerance values exceeded 0.1. This confirms that the independent variables are sufficiently independent, allowing for accurate estimation of their individual effects.

The level of significance was set as *p* < 0.05.

## 3. Results

### 3.1. Sample Characteristics

During the study period, 710 questionnaires were completed; forms with incorrect age groups, as well as incomplete and incorrect forms, were excluded. The mean age of involved mothers (*n* = 572) was 31.2 ± 4.2 years (min. 21–max. 46). The mean number of households was 3.3 ± 0.7 and the mean number of children in the household was 1.2 ± 0.5. The mean age of the children was 13.0 ± 4.8 months. Of the mothers, 67.3% had an adequate health literacy score. For children, 74.3% continued breastfeeding, and 30.6% had formula. As per the complementary feeding indicators 100%, 69.8%, 78.1%, and 58.9% of the children met ISSSF, MDD, MMF, and MAD, respectively. Table 1 describes the mothers’ and children’s sociodemographic characteristics and children’s complementary feeding indicators.

### 3.2. Associated Factors for Infant Young Child Feeding Indicators

Continued breastfeeding rate was associated with maternal age, mother’s education duration, and child’s age. Children aged 18–23 months were less likely to continue breastfeeding than those 6–8 months and mothers with age below 30 years and education duration more than 12 years are more likely to continue breastfeeding. Mother’s education duration and child’s age also showed significant difference in multivariate analysis (Table 2). The percentage of EFF was found to be higher when the mother’s age was over 30, the baby was older than 9 months, and the pregnancy duration was more than 38 weeks compared to other groups (*p* < 0.05). When maternal and infant characteristics were included in the multivariate logistic regression analysis, only maternal age and child age were found to be associated with the percentage of EFF. Univariate and binary analysis reveals that term infants and children aged 12–23 months were less likely to use bottles than those in the younger ages and preterm infants.

Children of mothers aged ≥ 30 years and had education > 12 years, older babies, and those whose household income was equivalent to outcome and higher were more likely to meet MDD and MAD. Children of working mothers were more likely to meet MDD, MMF, and MAD whereas the rate of MDD, MMF, and MAD was found to be lower in families with multiple children (Table 3).

Children aged 18–23 months [odds ratio [OR] = 24.84; 95% confidence interval [CI] = 3.08–200.05] and boys had higher odds of having sweet beverage consumption than those in the younger age groups and girls.

Children aged 18–23 months [odds ratio [OR] = 35.31; 95% confidence interval [CI] = 12.78–97.58] had higher odds of having unhealthy food consumption than those in the younger age groups. While UFC rates were found to be higher in the children of the mothers from the south, center and east, and from families with multiple children and term infants, the rate was found to be lower among children from mothers with education duration >12 years.

Children of mothers aged ≥ 30 years, from the west region, and with adequate TSOY-32 scores had lower odds of meeting zero fruit and vegetable consumption [odds ratio [OR] = 0.60; 95% confidence interval [CI] = 0.39–0.93]. Infants from families with multiple children and lower income were more likely to meet zero fruit and vegetable consumption than others. (Table 4)

## 4. Discussion

To the best of our knowledge, this was the first study to investigate complementary feeding practices, using the most recent WHO IYCF (2021) indicators with the participation of an adequate number of mothers from five regions of Turkey and examining correlation with mothers’ health literacy. In our study, the overall rates of continued breastfeeding and formula intake were 74.3% and 30.6%, respectively. While the breastfeeding rate was 80.0% among infants aged 6–8 months, it decreased to 56.1% among children aged 18–23 months. Based on the research, we have identified two significant characteristics that contribute to the continuation of breastfeeding: the younger age of the baby and the mother and the higher education of the mother [18,19,20].

In our study, all of the infants were having complementary feeding at 6–8 months. Of the infants, 38.8% were introduced to solid, semi-solid, and soft foods before 6 months and 61.2% at 6–8 months. According to the WHO reports published between 2010–2018, the complementary feeding indicator “ISSSF” was reported to be 63.1% in the world. It was found to be 66.4%, 58.2%, and 81.7% in low-income, middle-income, and high-income countries, respectively [18]. According to Turkey Demographic Health Survey data, the rate of ISSSF at 6 months was 85% in Turkey [21]. Senyazar et al. [22] revealed that the introduction rate of semi-solid soft foods before 6 months was 29.3%, the mean age of introduction was 5.6 months, and the common reason for early initiation was tasting in Turkey. In a previous study, Köksal et al. [23] revealed that two cities in the east of Turkey with high malnutrition prevalence had lower odds of early introduction of complementary feeding than the regions with low and medium malnutrition prevalence, but they introduced iron-rich foods later than the other regions. Several studies have evaluated regional, cultural, and ethnic differences in child feeding practices, which influence diet and access to nutritious foods [1,20,23,24,25]. One such study aimed to assess the breastfeeding and complementary feeding practices of mothers with children aged 12–23 months across three regions in Turkey, involving 1486 mother–child pairs [23]. Commonly introduced foods included yogurt, bread, fruits, and vegetables, while iron-rich foods like red meat, poultry, and fish were introduced later, particularly in regions with lower nutritional status. Another study examined maternal attitudes and children’s eating habits in thin and normal-weight children across two cities, Ankara and Şanlıurfa, which have distinct socioeconomic profiles [24]. It was found that mothers of thin children were more concerned about their children’s weight and applied more pressure to feed them. Differences in breastfeeding duration, formula use, and complementary feeding practices were observed between the two cities, highlighting the significant influence of sociocultural factors on feeding practices. Additionally, a study on infant feeding practices among Syrian refugee mothers, based on observations from Syrian healthcare workers (HCWs) in Turkey, revealed that many mothers used pre-lacteal sugary water and discontinued breastfeeding before 12 months. Barriers to breastfeeding included lack of education, mental and physical health challenges, food insecurity, and various sociocultural obstacles [25]. In our study, we could not find any difference in achieving MDD, MAD, MMF, or EFF (being iron-rich food) but we found a difference in ZVF between regions. Children of mothers from the west, where the Mediterranean diet is common, were less likely to meet ZVF than in the other regions. While families with multiple children had higher odds of ZVF than others, we found that children from low-income households ate less fruit and vegetables than high and equal-income households. To address these variations, educational programs or resources aimed at improving maternal health literacy should be customized and tailored to specific locations, focusing on child-feeding practices.

Since the WHO IYCF guidelines were published recently, we have limited data on complementary feeding indicators of the world. In a study conducted in the Middle East and North Africa region, MDD and MAD were found to be 38% and 21%, respectively [26]. Yunitasari et al. [27] reported MDD, MMF, and MAD from Indonesia to be 53.95%, 71.14%, and 28.13%, respectively. Naja et al. [28] reported MDD, MMF, and MAD from Lebanon to be 37.5%, 92.8%, and 34.4%, respectively. In our study, we found MDD, MMF, and MAD to be 69.8%, 78.1%, and 58.9%, with higher percentages than the other developing countries. In accordance with the previous studies, our multivariate analysis reveals that the older the mother and child, the higher the mother’s education year, the working status of the mother, and the higher household income had a greater effect on experiencing MDD and MAD [29,30,31]. We also found that while families with multiple children were less likely to achieve MMF, working mothers’ children had higher MMF percentages.

It was proven that children who consume eggs and flesh foods have higher intakes of various nutrients important for optimal linear growth [4]. As a new complementary feeding indicator in WHO IYCF, we found EFF at 87.1% in our study. We also found that the older infants and children from older mothers and term infants had higher odds of achieving EFF than others. Similar to our results, in the Lebanon study, it was shown that EFF increased with increasing maternal and infant age, and EFF decreased in those whose mothers were obese [28]. In our study, we found no significant relationship between household income and EFF.

Sweet beverage and unhealthy food consumption are two new complementary feeding indicators. Higher intakes of sugar-sweetened beverages was shown to be associated with an increased obesity risk among children of all ages and the early introduction of these beverages (before 12 months of age) is associated with obesity at six years of age [32]. Commercially prepared food products are often high in salt, sugar, and saturated fatty acids, energy-dense and nutrient-poor. As consumption of such foods may displace more nutritious foods, limit the intake of essential vitamins and minerals and be associated with lower length-for-age z-scores among 12–23 month old children, these products are not recommended for young children [33]. In our study, we found SwB at 5.6% and UFC at 21.9%. While we found that older infant age was the major factor associated with both SwB and UFC, boys were 2.8 times more likely to have sweet beverages. Our research also reveals that while UFC was elevated in term newborns and in households with numerous children, a greater degree of maternal education had a protective impact on UFC.

In our study, the education level of 90.6% of the mothers was more than 12 years and their TSOY-32 scores were 6.5% inadequate, 26.4.1% limited, 29.2% adequate, and 37.8% excellent, which were above the average of Turkey. Okyay et al. [16] found Turkish women’s TSOY-32 scores as 25.4% inadequate, 42.1% limited, 27.6% adequate, and 4.9% excellent [16]. When we investigated the relationship between complementary feeding indicators and TSOY-32 scores, we found that the only indicator that had an association with mothers’ TSOY-32 scores was zero fruit and vegetable consumption. The odds of zero fruit and vegetable consumption were 1.7 times lower in those with an adequate maternal TSOY score. This result may be due to the perception that a healthy diet is based on vegetables and fruits. If we raise awareness of the other complementary feeding indicators such as dietary diversity, meal frequency, egg and/or flesh food consumption, and the harms of unhealthy foods and sweet beverages, we will have a better complementary feeding status in IYCF and will find a relationship between TSOY-32 scores and the other complementary feeding indicators.

One limitation of our study is its descriptive, cross-sectional design, which limits the ability to establish cause-and-effect relationships. Additional studies employing experimental or intervention-based designs would be better suited to determine causal links.

A notable limitation of our study is the potential for selection bias. Mothers who, for various reasons—whether cultural, educational, or religious—do not actively participate in social media were automatically excluded from the research. This reliance on online recruitment and maternal self-reports may have introduced a skewed sample, as we predominantly reached individuals who are more engaged with digital platforms. Furthermore, the higher education level of the participants restricts the generalizability of the findings to the broader Turkish population, which includes a wider range of socioeconomic and educational backgrounds. Consequently, the results may not fully reflect the experiences of mothers from more diverse or less digitally connected communities. But it is possible that the responses themselves were not biased, as only motivated mothers were likely to participate. Given that both the educational level and TSOY-32 scores in this sample are above the national average, further research with a larger and more representative group of mothers is needed.

Another limitation of our study is the exclusion of children born from multiple pregnancies, which prevents us from providing insights into the feeding practices of twins or other multiples. Multiple pregnancies often come with distinct challenges that differ from singleton births [34,35], and without their inclusion, our findings cannot be generalized to this population. Additionally, incorporating multiple pregnancies would have required a larger sample size to ensure statistical validity, which was beyond the scope of this study. Future research should aim to include this group to provide a more comprehensive understanding of infant feeding practices.

A key strength of the study is the inclusion of mothers from five regions of Turkey, investigating complementary feeding practices, using the most recent multiple WHO IYCF-2021 indicators [4] with the participation of an adequate number of mothers, and examining the correlation these indicators with mothers’ health literacy.

## 5. Conclusions

The present study highlights that both maternal and child age are significantly associated with complementary feeding practices, specifically MDD, MAD, and EFF. Older mothers tend to feed their infants more promptly, while older infants show better adaptation to complementary feeding. Notably, having multiple children is the only independent variable that negatively impacts several complementary feeding indicators, including MDD, MMF, MAD, UFC, and ZVF. This suggests that children with more siblings may require closer monitoring and additional support to ensure adequate complementary feeding, and regular follow-ups may be necessary to prevent malnutrition.

Regional differences in complementary feeding practices were minimal, with the exception of ZVF. Interestingly, the only indicator associated with mothers’ TSOY-32 scores was zero fruit and vegetable consumption. This finding underscores the need for increased awareness of all complementary feeding indicators. By promoting greater awareness, we can improve overall complementary feeding practices within the framework of IYCF and potentially identify stronger associations between TSOY-32 scores and other complementary feeding indicators. These insights highlight the importance of targeted interventions and tailored education programs to enhance feeding practices and nutritional outcomes for infants and young children in Turkey.

## Figures and Tables

**Figure 1 nutrients-16-03537-f001:**
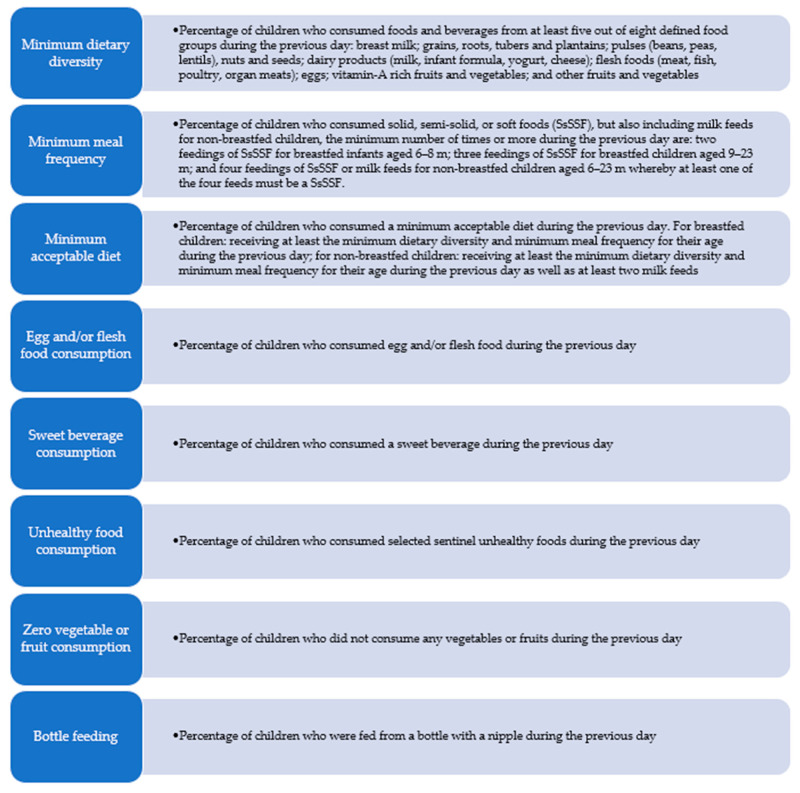
IYC complementary feeding indicators for children 6–23 months of age [4].

**Table 1 nutrients-16-03537-t001:** The sociodemographic characteristics of mothers and children; the complementary feeding indicators of the children, *n* = 572.

Family Characteristics	*n* (%) *	Child Characteristics	*n* (%) *
Maternal age ≥ 30 y	364 (63.6)	Child’s age	
Mother’s education > 12 y	518 (90.6)	6–8 m	125 (21.9)
Mothers having a job	408 (71.3)	9–11 m	121 (21.2)
Family structure, nuclear	521 (93.1)	12–17 m	203 (35.5)
Number of children ≥ 2	105 (18.4)	18–23 m	123 (21.5)
Region		Gestational duration ≥ 38 w	464 (81.1)
West	191 (33.4)	Infant sex, boy	320 (55.9)
South	84 (14.7)	Indicators	*n* (%) *
Center	159 (27.8)	Continued breastfeeding	425 (74.3)
North	76 (13.3)	Pacifier use	205 (35.8)
East	62 (10.8)	Bottle feeding	243 (42.5)
		Introduction of soft foods, 6–8 months	350 (61.2)
Household income		Formula feeding	175 (30.6)
Income < outcome	126 (22.0)	Minimum dietary diversity	399 (69.8)
Income = outcome	289 (50.5)	Minimum meal frequency	447 (78.1)
Income > outcome	157 (27.4)	Minimum acceptable diet	337 (58.9)
		Egg and/or flesh food consumption	498 (87.1)
Maternal TSOY-32 score		Zero vegetable/fruit consumption	124 (21.7)
Adequate	385 (67.3)	Unhealthy food consumption	125 (21.9)
Inadequate	187 (32.7)	Sweet beverage consumption	32 (5.6)

* Column percentage; TSOY: Turkish Health Literacy Scale.

**Table 2 nutrients-16-03537-t002:** The status of complementary feeding indicators [continued breastfeeding, egg and/or flesh food consumption, bottle-feeding] according to sociodemographic characteristics of mothers and children.

	Continued Breastfeeding				Egg and/or Flesh Food Consumption				Bottle-Feeding			
	% *	*p*	AOR	95%CI	% *	*p*	AOR	95%CI	% *	*p*	AOR	95%CI
**Overall**	74.3				87.1				42.5			
**Mother, age**												
<30 y	78.8	**<0.001**	**1.00**		**79.3**	**<0.001**	**1.00**		39.4	0.151	1.00	
≥30 y	71.7		**0.74**	0.47–1.76	**91.5**		**2.80**	**1.53–5.14**	44.2		1.44	0.97–2.14
**Mother, education**												
≤12 y	63.0	**0.036**	**1.00**		88.9	0.435	1.00		51.9	0.094	1.00	
>12 y	75.5		**2.05**	1.04–4.05	87.1		0.535	0.19–1.50	41.5		0.50	0.26–0.95
**Occupation**												
House wife	72.6	0.307	1.00		86.0	0.680	1.00		37.8	0.161	1.00	
Have a job	75.0		1.05	0.65–1.70	87.5		1.01	0.53–1.94	44.4		1.26	0.82–1.94
**Family structure**												
Nuclear family	74.7	0.314	0.74	0.65–2.50	87.5	0.199	1.77	0.75–4.16	42.2	0.400	0.94	0.50–1.74
Extended family	70.6		1.00		82.4		1.00		45.1		1.00	
**Region**												
West	70.7	0.108	1.00		88.5	0.771	1.00		45.5	0.360	1.00	
South	66.7		0.86	0.48–1.54	89.3		0.95	0.39–2.32	33.3		0.61	0.34–1.07
Center	80.5		1.74	1.03–2.95	84.3		0.98	0.49–1.96	40.9		0.78	0.50–1.23
North	77.6		1.55	0.79–2.99	86.8		1.27	0.52–3.08	46.1		0.89	0.50–1.56
East	75.8		1.39	0.69–2.76	87.1		0.84	0.33–2.16	45.2		0.88	0.48–1.61
**Number of children**												
Single	75.2	0.192	1.00		88.0	0.106	1.00		42.8	0.406	1.00	
>1	70.5		0.83	0.49–1.40	82.9		0.42	0.21–0.85	41.0		0.84	0.52–1.34
**Household income**												
In. < outcome	73.0	0.821	1.00		84.9	0.281	1.00		41.3	0.059	1.00	
In. = outcome	75.4		0.85	0.51–1.43	89.3		1.46	0.74–2.90	38.8		0.92	0.58–1.46
In. > outcome	73.2		0.23	0.38–1.25	84.7		1.02	0.48–2.16	50.3		1.34	0.80–2.25
**Mother’s TSOY-32**												
Inadequate score **	72.7	0.308	1.00		88.2	0.330	1.00		41.7	0.433	1.00	
Adequate score ***	75.1		1.25	0.82–1.90	86.5		0.93	0.49–1.46	42.9		1.02	0.70–1.48
**Infant’s age**												
6–8 m	**80.0 ^a^**	**<0.001**	1.00		**68.0 ^a^**	**<0.001**	**1.00**		**49.6 ^a^**	**0.006**	**1.00**	
9–11 m	**78.5 ^a^**		1.00	0.52–1.90	**90.9 ^b^**		**4.37**	**2.05–9.33**	**52.1 ^a^**		1.12	0.66–1.90
12–17 m	**79.3 ^a^**		1.08	0.60–1.95	**91.6 ^b^**		**4.41**	**2.27–8.56**	**36.5 ^b^**		**0.58**	**0.36–0.94**
18–23 m	**56.1 ^b^**		**0.34**	**0.18–0.61**	**95.1 ^b^**		**7.24**	**2.83–18.49**	**35.8 ^b^**		**0.56**	**0.33–0.97**
**Gestational week**												
<38 w	67.6	0.051	1.00		81.5	**0.043**	1.00		**56.5**	**<0.001**	**1.00**	
≥38 w	75.9		1.68	1.04–2.73	88.4		1.45	0.78–2.67	**39.2**		**0.51**	**0.33–0.79**
**Infant sex**												
Girl	73.8	0.443	1.00		87.7	0.393	1.00		44.0	0.279	1.00	
Boy	74.7		1.08	0.64–1.70	86.6		0.85	0.49–1.46	41.3		0.85	0.60–1.21

* Row percentage; In.: income; TSOY: Turkish Health Literacy Scale; ** inadequate and problematic health literacy groups were combined; *** adequate and excellent health literacy groups were combined; ^a,b^: different letters in the same column for the variables were significant, *p* < 0.05.

**Table 3 nutrients-16-03537-t003:** The status of complementary feeding indicators [minimum dietary diversity, minimum meal frequency, minimum acceptable diet] according to sociodemographic characteristics of mothers and children.

	Minimum Dietary Diversity				Minimum Meal Frequency				Minimum Acceptable Diet			
	% *	*p*	AOR	95%CI	% *	*p*	AOR	95%CI	% *	*p*	AOR	95%CI
**Overall**	69.8				78.1				58.9			
**Maternal age**												
<30 y	**59.6**	**<0.001**	**1.00**		76.9	0.332	1.00		**49.5**	**<0.001**	**1.00**	
≥30 y	**75.5**		**1.92**	**1.23–3.01**	78.8		1.12	0.70–1.80	**64.3**		**1.64**	**1.10–2.44**
**Mother’s education**												
≤12 y	**53.7**	**0.007**	1.00		72.2	0.174	1.00		**44.4**	**0.017**	1.00	
>12 y	**71.4**		1.34	0.68–2.65	78.8		0.98	0.48–2.00	**60.4**		1.08	0.56–2.06
**Occupation**												
House wife	**60.4**	**0.002**	1.00		**68.3**	**<0.001**	**1.00**		**45.7**	**<0.001**	**1.00**	
Have a job	**73.5**		1.47	0.92–2.36	**82.1**		**2.00**	**1.25–3.19**	**64.2**		**1.78**	**1.16–2.72**
**Family structure**												
Nuclear family	70.2	0.251	1.31	0.66–2.58	79.1	0.065	1.65	0.85–3.24	59.5	0.223	1.25	0.67–2.35
Extended family	64.7		1.00		68.6		1.00		52.9		1.00	
**Region**												
West	76.4	0.118	1.00		79.1	0.959	1.00		64.9	0.180	1.00	
South	70.2		0.69	0.36–1.31	79.8		1.06	0.55–2.06	59.5		0.80	0.45–1.41
Center	64.8		0.76	0.45–1.29	76.7		0.86	0.50–1.47	53.5		0.74	0.47–1.18
North	68.4		0.94	0.48–1.83	78.9		0.90	0.45–1.78	60.5		1.01	0.55–1.83
East	62.9		0.51	0.26–1.00	75.8		0.78	0.39–1.59	51.6		0.56	0.30–1.04
**Number of children**												
Single	**72.8**	**<0.001**	**1.00**		**80.1**	**0.014**	**1.00**		**61.9**	**0.002**	**1.00**	
>1	**56.2**		**0.36**	**0.22–0.61**	**69.5**		**0.55**	**0.33–0.93**	**45.7**		**0.45**	**0.28–0.73**
**Household income**												
In. < outcome	**58.7 ^a^**	**0.007**	1.00		80.2	0.273	1.00		**49.2 ^a^**	**0.025**	1.00	
In. = outcome	**74.0 ^b^**		**1.75**	**1.06–2.89**	75.4		0.64	0.37–1.18	**59.9 ^b^**		1.31	0.83–2.08
In. > outcome	**70.7 ^b^**		1.51	0.85–2.67	81.5		0.83	0.44–1.57	**65.0 ^b^**		1.65	0.97–2.80
**Maternal TSOY-32**												
Inadequate score **	68.4	0.352	1.00		77.0	0.360	1.00		58.3	0.451	1.00	
Adequate score ***	70.4		1.32	0.86–2.31	78.7		1.16	0.75–1.79	59.2		1.13	0.77–1.66
**Infant’s age**												
6–8 m	**40.8 ^a^**	**<0.001**	**1.00**		80.0	0.878	1.00		**38.4 ^a^**	**<0.001**	**1.00**	
9–11 m	**75.2 ^b^**		**4.48**	**2.50–8.04**	78.5		0.93	0.48–1.77	**62.8 ^b^**		**2.69**	**1.55–4.65**
12–17 m	**80.3 ^b^**		**5.97**	**3.47–10.25**	76.4		0.85	0.47–1.52	**66.0 ^b^**		**3.15**	**1.91–5.19**
18–23 m	**76.4 ^b^**		**4.72**	**2.60–8.58**	78.9		0.90	0.47–1.73	**64.2 ^b^**		**2.90**	**1.66–5.06**
**Gestational duration**												
<38 w	63.9	0.088	1.00		84.3	0.054	1.00		58.3	0.487	1.00	
≥38 w	71.1		1.16	0.71–1.91	76.7		0.63	0.35–1.12	59.1		0.91	0.57–1.44
**Infant sex**												
Girl	68.7	0.337	1.00		78.6	0.455	1.00		58.3		1.00	
Boy	70.6		1.07	0.71–1.60	77.8		0.96	0.63–1.45	59.4	0.434	1.03	0.72–1.48

* Row percentage; In.: income; TSOY: Turkish Health Literacy Scale; ** inadequate and problematic health literacy groups were combined; *** adequate and excellent health literacy groups were combined; ^a,b^: different letters in the same column for the variables were significant, *p* < 0.05.

**Table 4 nutrients-16-03537-t004:** The status of complementary feeding indicators [sweet beverage consumption, unhealthy food consumption, zero vegetable or fruit consumption] according to sociodemographic characteristics of mothers and children.

	Sweet Beverage Consumption				Unhealthy Food Consumption				Zero Vegetable or Fruit Consumption			
	% *	*p*	AOR	95%CI	% *	*p*	AOR	95%CI	% *	*p*	AOR	95%CI
**Overall**	5.6				21.9				21.7			
**Maternal age**												
<30 y	6.3	0.367	1.00		21.6	0.506	1.00		**26.9**	**0.015**	**1.00**	
≥30 y	5.2		0.47	0.19–1.13	22.0		0.53	0.31–0.90	**18.7**		**0.60**	**0.37–0.97**
**Mother’s education**												
≤12 y	11.1	0.070	1.00		**33.3**	**0.028**	1.00		29.6	0.097	1.00	
>12 y	5.0		0.56	0.18–1.75	**20.7**		0.58	0.26–1.30	20.8		0.95	0.46–1.92
**Occupation**												
House wife	6.7	0.546	1.00	0.42–2.50	23.8	0.503	1.00	0.73–2.22	28.7	0.013	1.00	0.43–1.13
Have a job	5.1		1.02		21.1		1.27		18.9		0.70	
**Family structure**												
Nuclear family	6.0	0.200	3.01	0.37–24.45	21.9	0.561	0.93	0.41–2.11	21.9	0.432	1.29	0.60–2.77
Extended family	2.0		1.00		21.6		1.00		19.6		1.00	
**Region**												
West	8.9	0.170	1.00		**16.2 ^a,b^**	**0.005**	**1.00**		**14.1 ^a^**	**0.011**	**1.00**	
South	4.8		0.46	0.14–1.51	**32.1 ^c^**		**2.27**	**1.15–4.48**	**25.0 ^b^**		**1.80**	0.92–3.51
Center	4.4		0.52	0.19–1.38	**24.5 ^b,c^**		**2.16**	**1.17–3.97**	**28.9 ^b^**		**2.19**	**1.26–3.83**
North	2.6		0.28	0.05–1.36	**13.2 ^a^**		0.95	0.40–2.25	**18.4 ^a,b^**		**1.33**	0.63–2.79
East	3.2		0.31	0.63–1.57	**29.0 ^c^**		**2.73**	**1.24–6.01**	**25.8 ^b^**		**2.14**	**1.04–4.42**
**Number of children**												
Single	5.4	0.369	1.00		**19.9**	**0.014**	**1.00**		**20.1**	**0.041**	**1.00**	
>1	6.7		1.71	0.62–4.72	**30.5**		**2.53**	**1.40–4.58**	**28.6**		**1.81**	**1.05–3.11**
**Household income**												
In. < outcome	7.9	0.410	1.00		26.2	0.409	1.00		**30.2 ^a^**	**0.015**	1.00	
In. = outcome	5.2		0.63	0.25–1.59	20.8		0.86	0.48–1.54	**21.1 ^b^**		0.67	0.40–1.12
In. > outcome	4.5		0.90	0.29–2.79	20.4		0.99	0.50–1.95	**15.9 ^b^**		**0.49**	**0.26–0.91**
**Maternal TSOY-32**												
Inadequate score **	3.2	0.058	1.00		18.7	0.123	1.00		**27.3**	**0.016**	1.00	
Adequate score ***	6.8		2.12	0.81–5.53	23.4		1.40	0.84–2.31	**19.0**		**0.60**	**0.39–0.93**
**Infant’s age**												
6–8 m	**0.8 ^a^**	**<0.001**	1.00		**4.0 ^a^**	**<0.001**	**1.00**		22.4	0.863	1.00	
9–11 m	**2.5 ^a,b^**		3.65	0.36–36.86	**7.4 ^a^**		**2.09**	0.66–6.60	23.1		1.14	0.60–2.15
12–17 m	**4.9 ^b^**		7.62	0.93–61.98	**24.1 ^b^**		**9.61**	**3.58–25.79**	19.7		0.94	0.52–1.70
18–23 m	**14.6 ^c^**		**24.84**	**3.08–200.0**	**50.4 ^c^**		**35.31**	**12.78–97.6**	22.8		1.12	0.59–2.14
**Gestational duration**												
<38 w	2.8	0.114	1.00		**13.9**	**0.015**	1.00		18.5	0.227	1.00	
≥38 w	6.3		1.70	0.48–6.00	**23.7**		1.95	0.98–3.86	22.4		1.31	0.75–2.29
**Infant sex**												
Girl	**2.8**	**0.007**	**1.00**		23.0	0.310	1.00		24.6	0.080	1.00	
Boy	**7.8**		**2.80**	**1.13–6.92**	20.9		0.88	0.55–1.41	19.4		0.78	0.51–1.18

* Row percentage; In.: income; TSOY: Turkish Health Literacy Scale; ** inadequate and problematic health literacy groups were combined; *** adequate and excellent health literacy groups were combined; ^a,b,c^: different letters in the same column for the variables were significant, *p* < 0.05.

## Data Availability

The raw data supporting the conclusions of this article will be made available by the authors on request.

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
