# Peer review of "Evaluation of Complementary Feeding Indicators Among Children Aged 6–23 Months According to the Health Literacy Status of Their Mothers"

_nutrients, 2024, doi:10.3390/nu16203537_

Round 1
Reviewer 1 Report
Comments and Suggestions for Authors
Dear Authors,
I found your article very interesting, and I am grateful for the opportunity to review it.
Here are the points that I believe should be better clarified:
- In 2.1. Study Design, you should provide reference(s) for the "snowball method"
- In 2.2. Study Population, please explain the rationale for excluding “Children born from multiple pregnancies.”
- Section 2.3. Variables: is there a translated copy of the questionnaire available to readers as a supplementary file? If not, I suggest you include it. The same applies to the TSOY-32
- Section 4. Discussion: when discussing the limitations and strenghts of the study, it should be acknoledged that the Study Design itself introduces a selection bias. Mothers who are not (for various reasons, whether cultural, educational, or religious) active on "social media" were automatically excluded from the research. Although you mention that "since the study was conducted online, it relied on maternal self-reports. Although this situation may suggest selection bias", and that "the higher education level of participants limits the generalizability of the results to the wider Turkish population", I believe these points need to be more explicitly explained
I think these are the main points that need to be addressed. Thank you very much
Author Response
R1.0. I found your article very interesting, and I am grateful for the opportunity to review it. Here are the points that I believe should be better clarified:
RR1.0. Thank you for your valuable comments. Please find our corrections regarding your comments below:
R1.1. - In 2.1. Study Design, you should provide reference(s) for the "snowball method"
RR1.1. The formal snowball method was used and a reference was added ” Gierczyk et al. 2023” and cited in the manuscript
R1.2. - In 2.2. Study Population, please explain the rationale for excluding “Children born from multiple pregnancies.”
RR1.2. In studies on infant feeding, children born from multiple pregnancies (e.g., twins, triplets) are often excluded to minimize potential confounding factors. Multiple pregnancies can result in unique physiological and developmental conditions, such as lower birth weights, preterm deliveries, or specific feeding challenges, which may not be representative of singleton births. These differences could skew the results and make it difficult to isolate the effects of feeding practices on infant outcomes. Additionally, to adequately represent multiple pregnancies, the sample size would need to be larger to ensure sufficient statistical power, which may not be feasible within the study's scope. By excluding these cases, the study ensures a more homogenous population, allowing for clearer conclusions about the relationship between infant feeding practices and health outcomes.
"... limitation of our study is the exclusion of children born from multiple pregnancies, which prevents us from providing insights into the feeding practices of twins or other multiples. Multiple pregnancies often come with distinct challenges that differ from singleton births [33,34], and without their inclusion, our findings cannot be generalized to this population. Additionally, incorporating multiple pregnancies would have required a larger sample size to ensure statistical validity, which was beyond the scope of this study. Future research should aim to include this group to provide a more comprehensive understanding of infant feeding practices.” Was added to the limitation section.
R1.3. - Section 2.3. Variables: is there a translated copy of the questionnaire available to readers as a supplementary file? If not, I suggest you include it. The same applies to the TSOY-32
RR1.3. Translated copy of the questionnaire and TSOY was given as Supplementary file.
R1.4. - Section 4. Discussion: when discussing the limitations and strengths of the study, it should be acknowledged that the Study Design itself introduces a selection bias. Mothers who are not (for various reasons, whether cultural, educational, or religious) active on "social media" were automatically excluded from the research. Although you mention that "since the study was conducted online, it relied on maternal self-reports. Although this situation may suggest selection bias", and that "the higher education level of participants limits the generalizability of the results to the wider Turkish population", I believe these points need to be more explicitly explained
RR1.4. We agree with this comment and “A notable limitation of our study is the potential for selection bias. Mothers who, for various reasons—whether cultural, educational, or religious—do not actively participate in social media were automatically excluded from the research. This reliance on online recruitment and maternal self-reports may have introduced a skewed sample, as we predominantly reached individuals who are more engaged with digital platforms. Furthermore, the higher education level of the participants restricts the generalizability of the findings to the broader Turkish population, which includes a wider range of socioeconomic and educational backgrounds. Consequently, the results may not fully reflect the experiences of mothers from more diverse or less digitally connected communities. Future studies should aim to incorporate a more varied recruitment strategy to ensure a more representative sample of the population.” Was added to the limitation section.
R1.5. I think these are the main points that need to be addressed. Thank you very much
RR1.5. Thank you for your valuable comments
Reviewer 2 Report
Comments and Suggestions for Authors
This is a very interesting study, however there are some problems:
- the bias of education: mothers are not representative for the whole population from this point of view.
- the bias of the sample: the number of mothers is VERY small for the population of Turkey
- the bias of being a longitudinal study
and many more. What can be done, taking in account that you cannot go back, is, from my point of view , to improve what you already have:
- maybe you have other sociodemographical data gathered (family support, father’s role in feeding, cultural beliefs about child nutrition, access to healthcare services), please use them, because they could provide a more comprehensive understanding of what affects complementary feeding practices.
- Including an assessment of how complementary feeding practices influence later childhood health (e.g., nutritional status, growth, cognitive development) would provide more actionable insights for healthcare providers and policymakers - at least discuss this aspect in the article
- discuss at least theoretically why certain regions perform better or worse in terms of feeding practices (e.g., differences in diet, access to nutritious foods, or local health policies). This can help to tailor interventions more effectively by region.
- again, also for discussions , what about interventions, such as educational programs or resources aimed at improving maternal health literacy.
Author Response
R2.0. This is a very interesting study, however there are some problems:
RR2.0. Thank you for your valuable comments. Please find our corrections regarding your comments below:
R2.1.- the bias of education: mothers are not representative for the whole population from this point of view
RR2.1. We agree with this comment and “The higher education level of mothers limits the generalizability of the results to the wider Turkish population.” was revised as “……, the higher education level of the participants restricts the generalizability of the findings to the broader Turkish population, which includes a wider range of socioeconomic and educational backgrounds.” in limitation section.
R2.2.- the bias of the sample: the number of mothers is VERY small for the population of Turkey
RR2.2. The sample size of the study was calculated and sufficient for our study purpose as seen in the method section; " Since there were no previous studies on health literacy and complementary feeding in the community, a preliminary study of 50 questionnaires was conducted to calculate the sample size. In this preliminary study we found the rate of minimum dietary diversity to be 75%. Previously, women’s health literacy (sufficient and excellent scores) was reported as 27.5-47.5% (mean: 35.5) in Turkey [8,15]. According to the health literacy status of the mothers, allocation ratio (n2/n1)=1.66 (64.5/35.5), p1=0.75, p2=0.60 (supposed 20% less in the group with inadequate health literacy), alpha error=0.05 for two tail, power %95, we calculated 551 mother-child pairs are necceassry to predict differences in minimum dietary diversity (G*Power 3.1.9.4, Franz Faul, Universitata Kiel, Germany). With a margin of error of 20%, the number of questionnaires to be reached was determined as 661.”
R2.3.- the bias of being a longitudinal study
RR2.3. We agree with this comment and “One limitation of our study is its descriptive design, which prevents the establishment of a cause-and-effect relationship. “ was revised as “One limitation of our study is its descriptive, cross-sectional design, which limits the ability to establish cause-and-effect relationships. Future studies employing experimental or intervention-based designs would be better suited to determine causal links”
R2.4. and many more. What can be done, taking in account that you cannot go back, is, from my point of view , to improve what you already have:- maybe you have other sociodemographical data gathered (family support, father’s role in feeding, cultural beliefs about child nutrition, access to healthcare services), please use them, because they could provide a more comprehensive understanding of what affects complementary feeding practices.
RR2.4. “Other sociodemographic factors, such as family support, the father's role in feeding, cultural beliefs about child nutrition, and access to healthcare services, may also influence complementary feeding practices. These factors should be considered, as they could provide a more comprehensive understanding of what affects feeding behaviors. It is important to take these variables into account when conducting research on complementary feeding to ensure a well-rounded analysis.” Was added.
R2.6. - Including an assessment of how complementary feeding practices influence later childhood health (e.g., nutritional status, growth, cognitive development) would provide more actionable insights for healthcare providers and policymakers - at least discuss this aspect in the article
RR2.6. “Understanding the association between maternal factors and IYCF practices could help policymakers design more effective public health strategies to support vulnerable populations.” was present in rationale of the manuscript. We have no data on how complementary feeding practices influence later childhood health (e.g., nutritional status, growth, cognitive development), and also our aim did not cover this subject therefore we could not discuss this subject in discussion.
R2.7. - discuss at least theoretically why certain regions perform better or worse in terms of feeding practices (e.g., differences in diet, access to nutritious foods, or local health policies). This can help to tailor interventions more effectively by region.
RR2.7. We agree with this comment and “Several studies have evaluated regional, cultural, and ethnic differences in child feeding practices, which influence diet and access to nutritious foods [1,19,22-24]. One such study aimed to assess the breastfeeding and complementary feeding practices of mothers with children aged 12–23 months across three regions in Turkey, involving 1,486 mother-child pairs[22]. Commonly introduced foods included yogurt, bread, fruits, and vegetables, while red meat, poultry, and fish were introduced later, particularly in regions with lower nutritional status. Another study examined maternal attitudes and children's eating habits in thin and normal-weight children across two cities, Ankara and Åžanlıurfa, which have distinct socioeconomic profiles[23]. It was found that mothers of thin children were more concerned about their child's weight and applied more pressure to feed them. Differences in breastfeeding duration, formula use, and complementary feeding practices were observed between the two cities, highlighting the significant influence of sociocultural factors on feeding practices. Additionally, a study on infant feeding practices among Syrian refugee mothers, based on observations from Syrian healthcare workers (HCWs) in Turkey, revealed that many mothers used prelacteal sugary water and discontinued breastfeeding before 12 months. Barriers to breastfeeding included lack of education, mental and physical health challenges, food insecurity, and various sociocultural obstacles[24].” Was added to discussion.
R2.8.- again, also for discussions , what about interventions, such as educational programs or resources aimed at improving maternal health literacy.
RR2.8. “To address these variations, educational programs or resources aimed at improving maternal health literacy should be customized and tailored to specific locations, focusing on child feeding practices.” Was added.
We sincerely thank you for your valuable contributions in making our study clearer and of higher quality.
Reviewer 3 Report
Comments and Suggestions for Authors
nutrients-3242233-peer-review-v1
This is interesting paper based on the survey and following analysis of the obtained answers. On one side this is simple paper, but authors have shown originality and way how to ask the questions and then to interpret the obtained results and to draw the conclusion. Well, from simple idea they have menage to build an interesting survey and to come will interesting, expected conclusions, but systematically analyzed.
Ln240: This reference needs to be according to the recommendations, as Senyazar et al. [19]
Ln256: This reference needs to be cited by number as Yunitasari et. [22]
Ln257: same as previous
References need to be formatted according to recommendations from the journal.
Author Response
R3.0. This is interesting paper based on the survey and following analysis of the obtained answers. On one side this is simple paper, but authors have shown originality and way how to ask the questions and then to interpret the obtained results and to draw the conclusion. Well, from simple idea they have menage to build an interesting survey and to come will interesting, expected conclusions, but systematically analyzed.
RR3.0. Thank you for your kind comments. Please find our corrections regarding your comments below:
R3.1. Ln240: This reference needs to be according to the recommendations, as Senyazar et al. [19]
RR3.1. corrected
R3.2. Ln256: This reference needs to be cited by number as Yunitasari et. [22]
RR3.2. corrected
R3.3. Ln257: same as previous
RR3.3. corrected
R3.4. References need to be formatted according to recommendations from the journal.
RR3.4. corrected
We sincerely thank you for your valuable contributions in making our study clearer and of higher quality.
Round 2
Reviewer 2 Report
Comments and Suggestions for Authors
I think that you answered well enough to all our observations and hope that in future studies you will also take into consideration the underlined limitations and correct them,